# Drug price dynamics following changes in procurement method in the public healthcare setting in Malaysia

Farahwahida Mohd Kasim[1,2], Ernieda Hatah[1]*, Lokhman Hakim Osman[3], Adliah Mhd Ali[1], Zaheer-Ud-Din Babar[4]

1 Faculty of Pharmacy, Centre for Quality Management of Medicines, Universiti Kebangsaan Malaysia, Wilayah Persekutuan Kuala Lumpur, Malaysia, 2 Pharmaceutical Services Programme, Ministry of Health, Petaling Jaya, Selangor, Malaysia, 3 Faculty of Economics and Management, Center for Value Creation and Human Well-being Studies, Universiti Kebangsaan Malaysia, Bangi, Selangor, Malaysia, 4 College of Pharmacy, Qatar University, Doha, Qatar

* ernieda@ukm.edu.my

## Abstract

Pooled and segmented procurement are common in public health, but changing needs may prompt method shifts. Understanding their impact on drug prices is crucial for informed policy decisions. This study aimed to evaluate the impact of procurement method changes from pooled to segmented (P→S) and from segmented to pooled (S→P) on drug prices within Malaysia's public health sector. A five-year retrospective analysis was conducted using data from eight therapeutic subgroups, covering 73 drugs and 151 comparative procurements. The product price index measured price changes between procurement groups, and differences in probabilities and price index were assessed using Pearson Chi-square and Mann-Whitney U tests. The significance level was $p < 0.05$. Out of 151 comparative procurements, 48.34% (n = 73) were P→S, and 51.66% (n = 78) were S→P. Significant differences were found in all pairwise comparisons between the groups for changes in drug prices. Specifically, the proportion of procurements with price decreases was significantly higher in the S→P group (79.49%) than in the P→S group (30.14%) ($p < 0.0005$), while price increases were more frequent in the P→S group (50.68% vs. 15.38%, $p < 0.0005$). Price stability was also more commonly observed in the P→S group (19.18%) than in the S→P group (5.13%) ($p = 0.008$). Overall, the P→S group had statistically significant higher mean ranks of 101.08 than the S→P group (mean rank = 52.53) with U = 151, z = −6.824, $p < 0.0005$, indicating that the S→P group tended to have lower prices on average than the P→S group. Pooled procurement potentially saves costs over segmented methods, but understanding both options' pros and cons is crucial for tailored solutions.

**Data availability statement:** All relevant data are within the paper and its Supporting Information files. Additional data that support the findings of this study are available upon reasonable request from the corresponding author, subject to ethical and legal restrictions.

**Funding:** This study was funded by the Fundamental Research Grants Scheme of the Ministry of Higher Education of Malaysia. Grant number FRGS/1/2022/SS02/UKM/02/1 was assigned, and EH received this funding. The funder had no role in the study design, data collection and analysis, decision to publish, or preparation of the manuscript. Funder website: https://mastic.mosti.gov.my/sti/incentives/fundamental-research-grant-scheme-frgs.

**Competing interests:** The authors have declared that no competing interests exist.

## Introduction

Pharmaceuticals play a pivotal role in enhancing health outcomes, fostering economic growth, and often mitigating the demand for other health resources by alleviating certain costs [1]. Despite these benefits, global health expenditure has nearly doubled, rising from $4.2 trillion in 2000 to $8.5 trillion in 2019, which is equivalent to 9.8% of the world's gross domestic product (GDP) [2]. There has been a notable upward trend in government total health spending over the past two decades, particularly in upper-middle and high-income countries [2]. In addition, projections for global spending on drugs indicate a continued rise, with expectations of a 3–6% increase, ultimately reaching a staggering $1.6 trillion by 2025 [3]. Similarly, in Malaysia, total health expenditure in nominal terms surged by 651.6%, increasing from RM8,556 million in 1997 to RM64,306 million in 2019 [4]. This rise includes a notable increase in out-of-pocket health expenditure, which grew from RM3,166 million in 1997 to RM22,492 million in 2019 [4]. A key trend within this growth is the upward trajectory of the Ministry of Health (MOH)'s drug expenditure, which observed a 21.1% increase from RM2,107.61 million in 2016 to RM2,552.18 million in 2021 [5,6]. This surge has placed considerable strain on the government budget, prompting the inclusion of cost-containment strategies in the National Medicines Policy to ensure the sustainability of healthcare services [7]. In this context, adopting efficient and cost-effective procurement strategies has become increasingly crucial to curb the growing pressure of drug-related expenditures.

Pharmaceutical procurement strategies can generally be categorized as collaborative or non-collaborative. Collaborative procurement refers to joint purchasing efforts among multiple facilities or agencies, including mechanisms such as pooled procurement and joint price negotiations, which are believed to help achieve more competitive pricing and improve accessibility [8,9]. Pooled procurement, in particular, may reduce prices, enhance bargaining power, and potentially stimulate supplier competition by leveraging aggregated demand and possible economies of scale. However, it is crucial to recognize that the success of pooled procurement collaboration is not guaranteed and can be influenced by various factors. Despite the generally positive outcomes observed in many studies after the implementation of pooled or centralized procurement, it was observed that there is a potential bias in the existing literature [9,10]. Most papers typically highlight instances of successfully implemented procurement practices, creating gaps in our understanding of initiatives, especially those that may not progress to the procurement stage or those that falter after being put into practice [10]. This bias in the literature raises questions about the comprehensiveness of our knowledge and emphasizes the importance of exploring both successful and unsuccessful cases regarding pooled procurement strategies. In contrast, non-collaborative procurement refers to purchasing conducted independently by individual facilities or agencies. This approach may offer greater autonomy and flexibility, as it allows hospitals, clinics, or departments to determine their procurement decisions based on specific needs, preferences, and priorities. Common forms include decentralized or segmented purchasing. While it may not benefit from the aggregated demand advantages seen in collaborative approaches, its notable strengths lie in the simplicity, ease, and flexibility of implementation.

Like many other countries, Malaysia's public healthcare sector employs both collaborative and non-collaborative procurement strategies for drug acquisition, balancing the benefits of consolidated purchasing power with the flexibility to meet specific institutional needs. Under collaborative procurement strategies, Malaysia adopts a pooled procurement system that leverages demand aggregation. This system includes centralized contracts and concession agreements, categorized as pooled procurement. The pooling involves consolidating quantities, either exclusively within MOH facilities or across multiple government entities, including the Ministry of Defense and the Ministry of Higher Education, to optimize purchasing power. In this system, the Malaysian MOH headquarters serves as the contract administrator, while public health facilities act as the purchasers of these contracts. Centralized contracts involved open tenders or negotiated tenders (in cases involving sole-source suppliers) between the government and supplier companies. Items were tendered when a drug's procurement value was expected to reach RM500,000 or more per year in one of the MOH's health facilities [11]. These central contracts were typically awarded to bid-winning companies for a duration of two or three years and were bound to a contracted quantity. Another key component of Malaysia's pooled procurement approach is the concession contract, also known as the Approved Products Purchase List. Unlike centralized contracts, the concession system allows a concessionaire company to procure and distribute medicines listed under the Approved Products Purchase List to public healthcare facilities. Once awarded, the concessionaire gains exclusive rights to supply a wide range of medicines to the public healthcare sector for a specified period. The pooled procurement system is designed to streamline procurement processes, ensuring a consistent supply of drugs at predictable costs, and is aligned with specific national procurement objectives, such as ensuring supply security and resilience within the public healthcare system.

On the other hand, under non-collaborative strategies, Malaysia employs a segmented procurement system, where individual public health facilities are responsible for managing the entire procurement process. This includes the direct purchase of drugs, where each facility is allowed to procure a single drug, strictly adhering to the MOH's Medicines Formulary list, valued at under RM50,000 per year. In these cases, a minimum of three (3) suppliers, registered with the Ministry of Finance, must be invited to submit quotations, ensuring competitive pricing [11]. For a drug valued between RM50,000 and RM500,000 per year per facility, a specific process known as "Lampiran Q" is applied, which requires the invitation of at least five quotations to allow more competitors to participate in higher-value procurement [11]. An exception to this is the national hospital, which serves as a tertiary referral center and is allowed a higher limit for *Lampiran Q*, with a threshold of up to RM1 million [11]. The procurement process for *Lampiran Q* is more complex, as it involves multiple committees, including the Specification Committee, Quotation Opening Committee, Technical Evaluation Committee, Financial Evaluation Committee, and the Selection Committee [11]. This detailed process does not apply to direct-purchase items. In the case of staggered procurement, a local contract is also required for *Lampiran Q*, but it is solely for individual facility agreements, tailored to meet the specific procurement needs of each facility.

Although the procurement strategies highlight distinct processes, the drugs can be dynamically switched between these strategies due to the varying thresholds and conditions applied. Drugs purchased under segmented procurement can be transitioned to pooled procurement methods when the purchase value by public healthcare facilities is expected to exceed the threshold limit within the financial year. This transition may occur when an item is first introduced to a centralized contract. Additionally, this may occur after segmented procurement has been conducted prior to the effective date of the new contract, typically due to lapses in the previous contract or delays in tender renewals. On the other hand, pooled procurement may transition to segmented procurement after the expiration of a centralized contract, particularly if a new tender is not awarded in time, the contract is terminated early due to fulfillment of quantities, or the MOH headquarters decides not to renew the contract upon its expiration.

Despite the streamlined processes across public healthcare facilities governed by Malaysia's Treasury and guided by standardized procurement guidelines [11], drug prices in the Malaysian public healthcare system remain high compared to international price standards [12,13]. On average, the prices in Malaysia's public procurement system, which includes MOH hospitals and clinics, and three university hospitals under the Ministry of Higher Education were reported to be

between two to three times the international reference prices [13]. Additionally, this study found that centrally managed tenders in Malaysian public pooled procurement, do not appear to result in prices that are significantly lower than those obtained through segmented purchasing of facilities when compared to the international reference prices [13]. Recognizing potential challenges in both collaborative and non-collaborative procurement strategies underscore the importance of understanding their nuances. This understanding significantly influences an organization's ability to achieve cost-effectiveness and sustainability in the acquisition of pharmaceuticals. By acknowledging procurement as a critical function and distinguishing between collaborative and non-collaborative strategies, organizations can strategically align their efforts to address the escalating drug spending and enhance equitable access to essential medicines. Given the dynamic nature of the public procurement method, allowing shifts between pooled and segmented approaches, it is imperative to evaluate the impact on drug prices during these transitions, a dimension that is currently unexplored in public health contexts. Therefore, this study aims to evaluate the impact of changing the procurement methods between pooled and segmented approaches on the cost dynamics within Malaysia's MOH. The anticipated findings are expected to offer valuable insights that inform more strategic and cost-effective procurement practices.

## Materials and methods

### Study design

This study is a retrospective observational analysis of database records pertaining to the pooled and segmented procurement of pharmaceutical products within Malaysia's public healthcare system. The ethical approval was obtained from the Medical Research and Ethics Committee, MOH (NMMR ID: 22-01094-9CC (IIR)) and from the Universiti Kebangsaan Malaysia Research Ethics Committee (UKM PPI/111/8/JEP-2022–753). Drug procurement data from the selected MOH facilities were collected between September 2022 and December 2023.

### Data sources and selection

The workflow of the data collection and selection process is shown in Fig 1. A purposive selection was made, focusing on therapeutic subgroups of drugs with notable spending and consumption patterns over the last five years. Specifically, eight therapeutic subgroups stood out for their significant expenditure and consumption within the Malaysian MOH in 2016 [14]. These included drugs for diabetes (A10), antithrombotics (B01), agents acting on the renin-angiotensin system (C09), lipid modifiers (C10), antibiotics for systemic use (J01), antineoplastic agents (L01), immunosuppressants (L04), and psycholeptics (N05). The therapeutic subgroups refer to the second level of the Anatomical Therapeutic Chemical classification system. The MOH Medicines Formulary list typically includes detailed specifications for each medicine, such as active ingredients, strengths, and dosage forms. From the list of drugs within these eight therapeutic subgroups (n = 420), data were further included in the analysis if the annual expenditure at the Anatomical Therapeutic Chemical classification level five was RM4 million or more in the public sector (IQVIA databases), reducing the sample to n = 159. Focusing on high-expenditure drugs allowed for sufficient data representation across both pooled and segmented procurement strategies to support meaningful comparison. This study included only drugs procured under both pooled and segmented procurement methods between 2017 and 2021 (a 5-year period). Pooled procurement through concession contracts was excluded, as these drugs are typically governed by long-term agreements with fixed procurement arrangements, making them unsuitable for price comparison across procurement modes. For segmented procurement, products were included if they had been quoted and purchased at least once by one of the selected healthcare facilities.

### Data collection process

The pooled procurement data was systematically gathered from the contract databases maintained by the Pharmaceutical Services Programme, MOH. This division oversees and facilitates drug-pooled procurement processes at

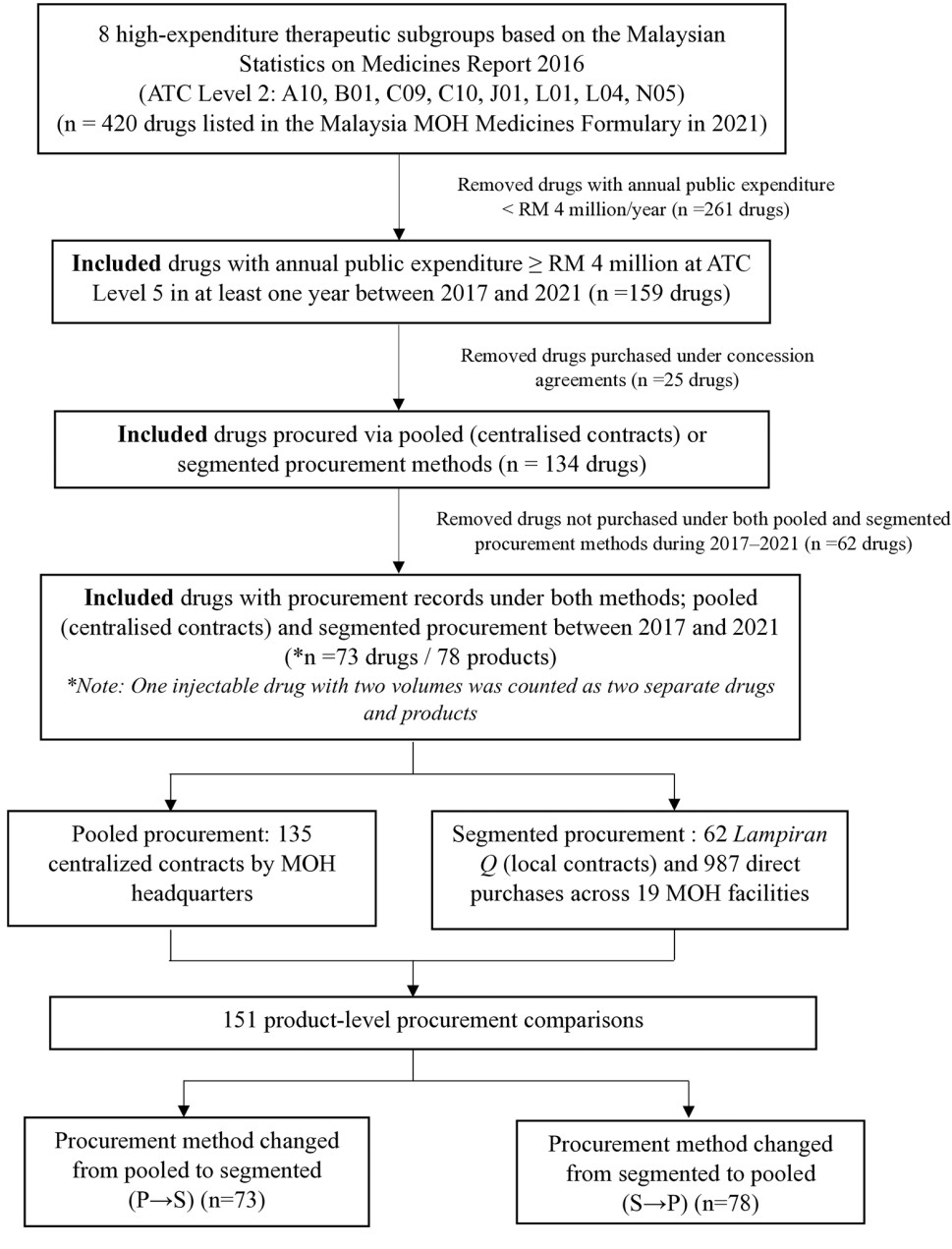

**Fig 1. The workflow of the data collection and selection process.**

the headquarters level, ensuring centralized coordination across MOH facilities. The data gathered were the bidding year, contract number, contract effective date, purchase price, active ingredients, product brand name, strength, dosage form, quantity, and pack size of the awarded contracts. An award signifies the official notification to a bidder that they have been selected as the successful party for a specific contract, valid for a predetermined period. On the other hand, segmented procurement or facility-based procurement data encompass *Lampiran Q* and direct purchases, collected from 19 MOH facilities, including hospitals, health clinics, and central stores, spread across five geographical zones - north, central, south, and east coasts in Peninsular Malaysia and East Malaysia. The strategic selection

of these facilities was based on their status as some of the highest spenders on drugs within the MOH. The collected dataset includes essential details such as the quotation number, quotation contract effective date, quotation date, local purchase order number, date of purchase, active ingredients, product brand name, strength, dosage form, purchase price, quantity, and pack size. To ensure data accuracy, procurement records were cross-checked with the facility's Pharmacy Information System (PhIS) and validated against the national eProcurement System, a government-mandated platform for all procurement transactions. The eProcurement System serves as the official reference for verifying purchase prices through supplier-issued purchase orders. As a real-time transaction-based system, missing data or discrepancies are uncommon. However, if any inconsistencies were found between the two systems, the responsible officer was contacted to clarify and resolve the issue, ensuring data reliability. All drug prices were collected and documented in Malaysian Ringgit (RM) to ensure a fair comparison over time, as all purchases were consistently conducted in RM.

After mapping, a smaller number of drugs were identified as being procured under both pooled and segmented procurement methods between 2017 and 2021, narrowing the sample to n = 73. Among the 73 drugs that met the inclusion criteria, a total of 78 products, referring to specific brand names and strength (e.g., 30 mg) and dosage form (e.g., modified-release tablets), were included in the final analysis. Within this selection, there were 135 centralized contracts (123 contracts were pooled within MOH facilities, while 12 contracts were pooled among the three ministries), 62 instances of *Lampiran Q*, and 987 direct purchases. It is important to note that these figures may represent multiple acquisitions of a brand at different timeframes. Each comparative dataset was analyzed for each switch that occurred at a specific time. If a product switched more than once at different timeframes, the data was treated independently and presented as separate datasets for each switch. To examine price changes from pooled to segmented (P→S) procurement, 73 comparative procurements were identified and assigned to this group. Similarly, a total of 78 comparative procurements were identified for the procurement method change from segmented to pooled (S→P). Thus, the study recognized a total of 151 comparative procurements, involving switching between pooled and segmented procurement methods, and vice versa.

## Data analyses

The product price index method was used to measure the price changes between the two procurement change groups. Relative variations in the price were measured using the simple price index [15]. Base and observation drug prices were established for the same brand and its identical attributes at uniform packaging. This procedure ensures that price comparisons between different procurement methods are consistent and fair. Median price was used as a summary statistic to represent segmented product price, while pooled procurement used a single data point. When certain subsets of segmented data have only one purchase, their median is determined solely by the product's purchased price, providing a robust measure despite the limited data points. The base price index for each product was set at 100, indicating that the index would remain at 100 in the absence of any price changes. However, if there is a price increase during the observation period, the price index would be more than 100. If there was a price decrease during the observation period, the price index would be less than 100. The measurements of price changes for a product or brand within the procurement change groups are as follows:

i)   P→S group

$$Pi_a = \frac{Pi,t}{Pi,0} \times 100$$

$Pi_a$,t = Median purchase price at observation period (segmented procurement)
$Pi_a$,0 = Central contract price at base period (pooled procurement)

ii)  S→P group

$$Pi_b = \frac{Pi,t}{Pi,0} \times 100$$

$Pi_b,t$ = Central contract price at observation period (pooled procurement)
$Pi_b,0$ = Median purchase price at base period (segmented procurement)

The analysis was conducted to elucidate the outcomes of the price effect after the change in procurement mode. First, a comprehensive descriptive analysis was employed to unveil the data characteristics. Then, the price changes that serve as the dependent variables, were categorized into three distinct classes: "Decreased," "Increased," and "No Change." The independent groups were the P→S group and the S→P group. Then, the chi-square test was used to determine the differences in the probabilities of price changes between these two independent groups. A post-hoc test using the z-test of two proportions was then conducted to assess differences in the proportions of the dependent and independent categories. To control for multiple comparisons, the Bonferroni adjustment was applied by dividing the alpha level of 0.05 by the number of pairwise comparisons, resulting in an adjusted alpha level of 0.016667 for three pairwise comparisons. Subsequently, the analysis aimed to determine if there were differences between the two independent groups, S→P and P→S, concerning the price index. Descriptive statistics, including percentage, minimum, maximum, median, and interquartile range of the price changes, were used to describe the characteristics of the procurement groups data. Given the non-normally distributed data, a Mann-Whitney U test, a non-parametric test, was employed to analyze the differences in the price index. Visual inspection was utilized to ascertain whether the distributions of the price index for both groups were comparable or not. The significance level was set at a p-value of less than 0.05. Data was analyzed descriptively and inferentially using the IBM Statistical Package for Social Science (SPSS®) software version 25.0. The workflow of the data analysis process is presented in Fig 2.

Subgroup analyses were conducted to offer a more comprehensive description of procurement activities, encompassing the types of products purchased, the types of competing products and the types of competing agents within each procurement change group. Within the S→P group, an analysis of the types of products purchased was performed based on tender types, distinguishing between first-time awards and subsequent tenders. These analyses were presented in a descriptive format to offer additional insights into their characteristics. Data was analyzed descriptively using the Microsoft® Excel® for Microsoft 365 MSO (Version 2312).

## Results

A total of 73 drugs from eight selected therapeutic subgroups were acquired through pooled and segmented procurement between 2017 and 2021. Among the drugs available, there are 78 products on the market from 29 licensees, managed by 25 government suppliers. In this scenario, 10 suppliers successfully secured contracts through pooled procurement, while 22 suppliers were successfully selected through segmented procurement. Interestingly, among the 25 government suppliers, only 28% (n = 7 suppliers) had their products purchased from pooled and segmented procurement methods. Of the 73 drugs, a total of 151 comparative product procurements were identified to change their procurement methods which is 73 (48.34%) from pooled to segmented procurement (P→S), and 78 (51.66%) had a change from segmented to pooled procurement (S→P). These drugs belonged to the therapeutic subgroup of N05 (23.84%, n = 36), L04 (18.54%, n = 28), B01 (17.88%, n = 27), L01 (11.26%, n = 17), C09 (8.61%, n = 13), A10 (7.95%, n = 12), C10 (6.62%, n = 10), and J01 (5.30%, n = 8). Samples data were mainly originator and reference of biologics drugs (79.47%, n = 120). Local and imported generics accounted for a similar percentage of 9.93% (n = 15) each and biosimilars accounted for the smallest proportion of 0.66% (n = 1) of the total procurement. The summary of therapeutic subgroups and types of products according to the procurement groups are presented in Figs 3 and 4. Additionally, the quantities awarded through pooled procurement were

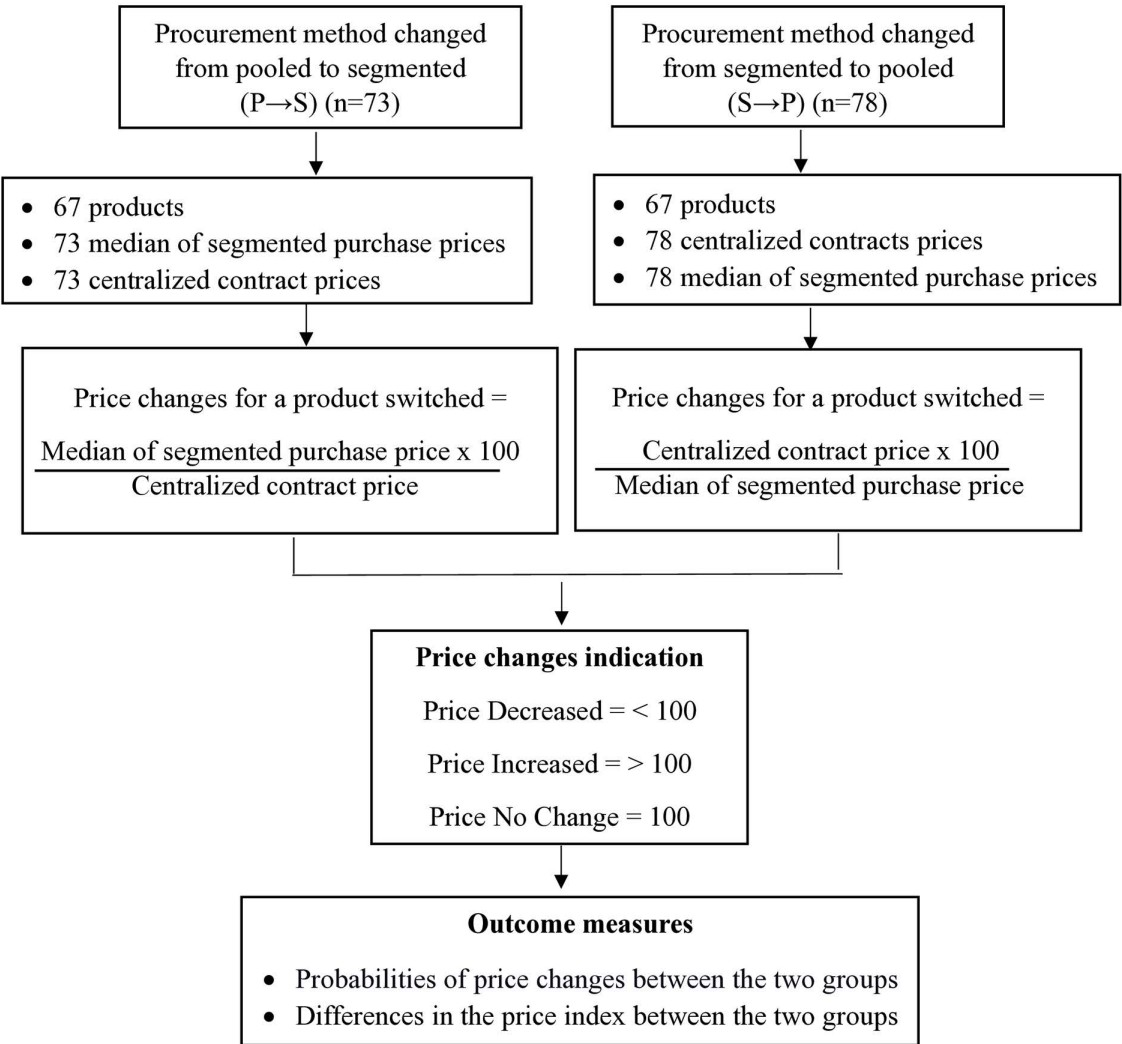

**Fig 2. The workflow of the data analysis process.**

higher than those purchased through segmented procurement during the transition period. Overall, the volume ratio in the P→S transition ranged from 12:1–66,000:1, while in the S→P transition, it ranged from 1:25–1:135,000.

The proportion of the P→S group that has price decreased was significantly different from the proportion of the S→P group (30.14%, n = 22 versus 79.49%, n = 62), with a p-value of < 0.0005. There was also a statistically significant increase in price in the proportion of the P→S group versus the S→P group (50.68%, n = 37 versus 15.38%, n = 12) with p < 0.0005. Price remained unchanged significantly in the P→S group than the S→P group (19.18%, n = 14 versus 5.13%, n = 4) with p = 0.008. The summary of procurement changes and their effect on price are presented in Fig 5. In the P→S group, the median price index is 100.18 (IQR: 107.11 - 99.34) with the minimum and maximum price indexes of 48.62 and 311.17, respectively. In contrast, the S→P group has a median price index of 95.13 (IQR: 99.62 - 88.82) with a minimum and maximum price index of 46.54 and 139.66, respectively. Further analysis found that there are significant differences in terms of price lowering effect between the P→S group (mean rank = 101.08) and the S→P group (mean rank = 52.53), U = 151, z = −6.824, p = < 0.0005. The results show that the

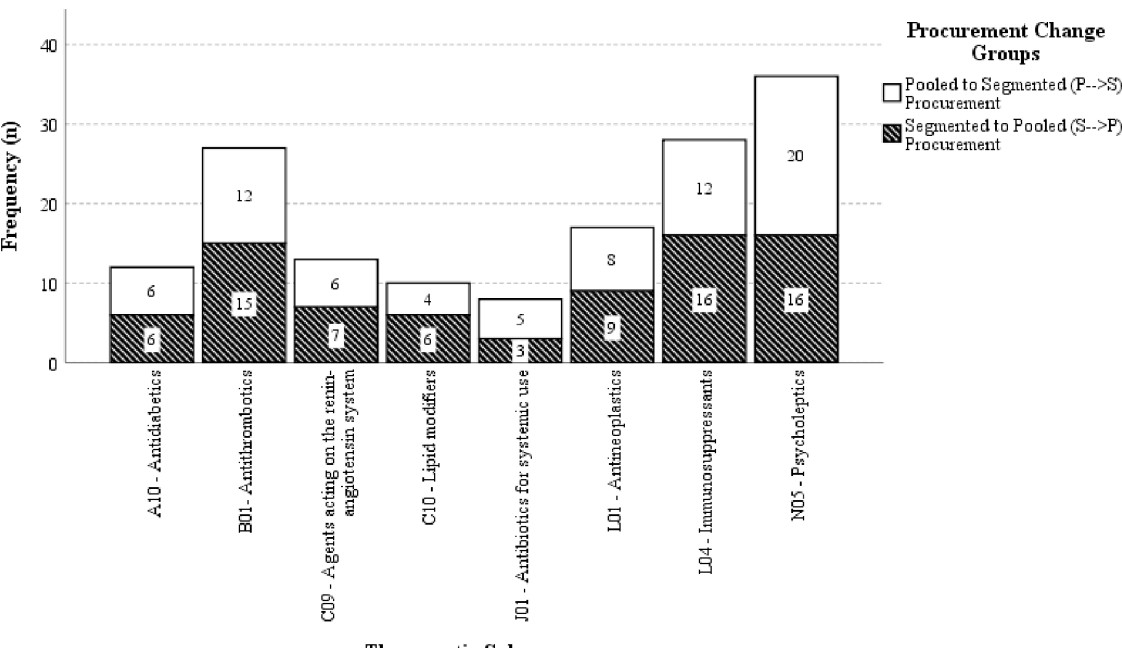

**Fig 3. The distribution of the samples among the procurement changes groups and the therapeutic subgroups.**

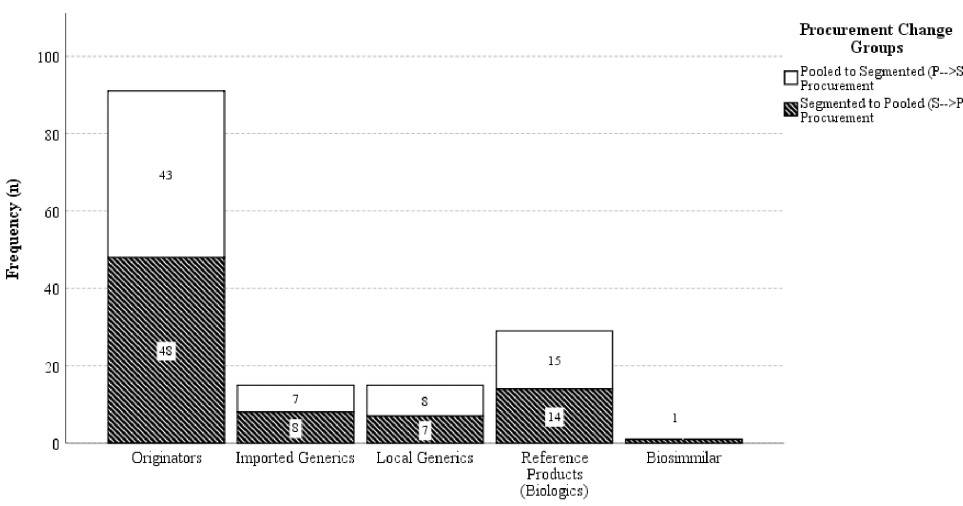

**Fig 4. A visual representation of the distribution of the samples across the groups of procurement change and product types.**

S→P group tends to have an average of lower prices than the P→S group. Fluctuations in drug prices observed during procurement changes are presented in Fig 6.

Further analysis focusing on the price effects within the P→S group related to types of products purchased which are originator or reference versus generic, exhibited a varied trend of price decreases, increases, or remaining the same.

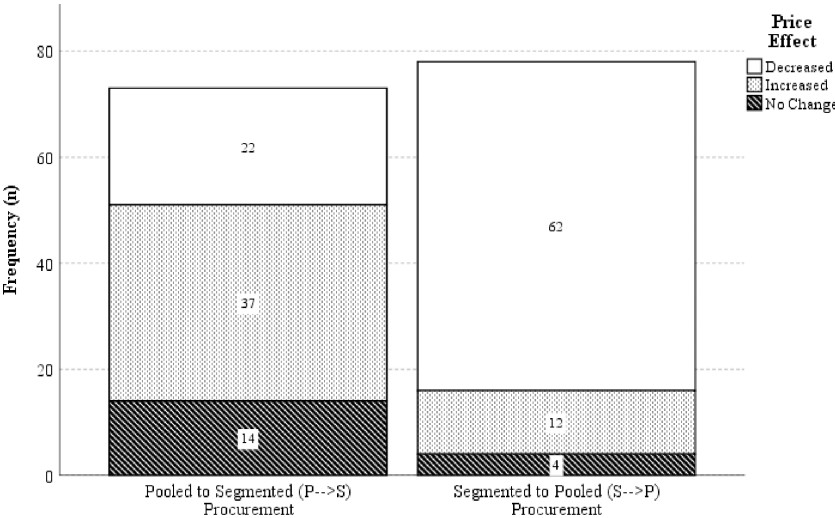

**Fig 5. The effect of the drug price on the procurement change groups.**

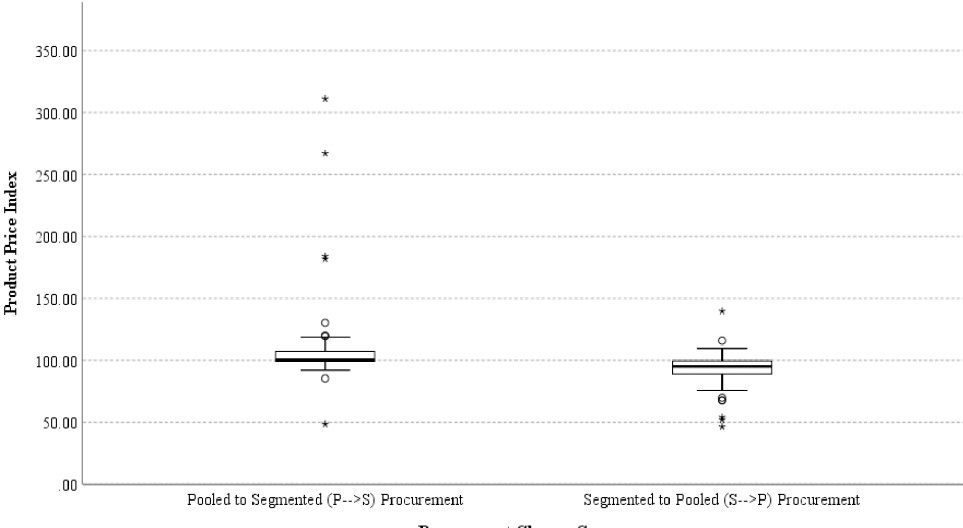

**Fig 6. The fluctuations in drug prices during procurement change.**

However, a subtle deviation in trends was noticed among generic products (26.67%, n = 4) when switching from P→S in which less numbers of drugs have prices decreased than the originator or reference biologic product groups (31.03%, n = 18). While the originator or reference biologic product groups tend to have more price increases (53.45%, n = 31) versus generics (40.00%, n = 6) in the P→S group. Notably, only three purchases were made under segmented procurement after existing contracts had expired and were not renewed, resulting in a price increase in these instances. On the contrary, prices that remained unchanged in generics were marginally higher in the P→S group compared to those in the originator or reference biologic product groups (33.33%, n = 5 versus 15.52%, n = 9). In the initial tendering phase S→P

group, most of the drugs exhibited a price decrease. However, as subsequent tenders progressed, there was a slight decline in the degree of price reduction for both originator or reference biologic products (initial: 85.71%, n = 12 versus subsequent: 77.08%, n = 37) and generic or biosimilar products (initial: 87.50%, n = 7 versus subsequent: 75.00%, n = 6).

Price transition was also analyzed based on the types of competing products and supplying agents. Single or mixed competing products and agents were evaluated using both the observation and base data across the two procurement groups. Single competing product or single agent refers to the participation of one product or agent in both the observation and base data periods. Mixed competing products refer to the involvement of multiple products or a mix of single and multiple products in the competition. Similarly, mixed competing agents indicate the participation of multiple agents or a combination of single and multiple agents in the competition. It is important to note that a single competing product could be represented by one or multiple suppliers offering the same product during the bidding process. In the P→S group, single competing product show a relatively smaller decrease in prices than mixed competing products (21.28%, n = 10 versus 46.15%, n = 12) and a more moderate increase in prices (63.83%, n = 30 versus 26.92%, n = 7). In this P→S group, pooled procurement specifically pertains to the quantities pooled within MOH facilities. Conversely, in the S→P group, the price trends between single competing product and mixed competing products are mostly similar between increase, decrease or remain. In the S→P group, a substantial trend of price decrease was observed, particularly when quantities were pooled between the MOH, Ministry of Defense, and Ministry of Higher Education, with 91.67% (n = 11 out of 12) of purchases demonstrating a price decrease. Notably, within this procurement group, five purchases involved a single competing product, while six purchases involved mixed competing products. This contrasted with quantities pooled solely within the MOH, where only 77.27% (n = 51 out of 66) showed a reduction in price, regardless of the types of competing products. In addition, the types of competing agents remained largely consistent with the overall findings. In the P→S transition, only three products had a single supplier participating in both procurement phases, one of which experienced price reductions. This finding supports the potential explanation for price decreases in some P→S products, as the majority of these products involved more than one agent. The summary of further analyses on the types of purchased products, competing products, and competing agents is tabulated in Table 1. Details of price changes in the P→S and S→P procurement groups are provided as supplementary information in S1 Appendix.

## Discussion

This study delves into the dynamics of drug prices when the procurement method shifts between pooled and segmented approaches in the public healthcare system in Malaysia. Overall, drug prices offered by suppliers either decreased, increased, or did not change after the change in procurement methods in this study. It shows that procurement methods in the public health care system do allow for dynamic pricing. The initial study's findings illuminate a notable variation in supplier involvement between segmented and pooled procurement methods, signifying a diverse procurement landscape within the therapeutic subgroups. This variance underscores the adaptability and versatility of the selected suppliers, showcasing their capability to navigate and engage across different procurement frameworks and it is intriguing to observe that only a minority of suppliers demonstrate the capacity to operate seamlessly within both pooled and segmented procurement frameworks, potentially contributing to the streamlining of the acquisition process.

The key findings emphasize that transitioning procurement methods may provide a significant price advantage, effectively resulting in reduced drug prices for Malaysia's public sector. The present study reveals a significant decrease in prices within the S→P group when compared to the P→S group. This finding aligns with similar observations in other studies conducted in locations such as South Africa [16] and seven low- to middle-income countries [17]. These studies highlight that drug costs can be effectively reduced through strategies like tendering, centralized, or pooled procurement. The initial decline in prices observed in the S→P group, leading to enhanced cost efficiency, can be recognized to several complementary factors. Firstly, public buyers within this group may possess greater negotiating leverage, allowing them to secure more favorable pricing conditions. Negotiations frequently take place in the context of pooled procurement,

**Table 1. The impact of prices is analyzed based on the types of purchased products, competing products, and competing agents.**

| Subgroup Analyses | Procurement Changes Groups | Variables | n | Price Decreased | Price Increased | Price Remain |
|---|---|---|---|---|---|---|
| **Types of purchased products** | P→S procurement group | Originators and reference of biologic products | 58 | 31.03% (n=18) | 53.45% (n=31) | 15.52% (n=9) |
| | | Generic products | 15 | 26.67% (n=4) | 40.00% (n=6) | 33.33% (n=5) |
| | S→P procurement group | Originators and reference of biologic products | 14 | Initial: 85.71% (n=12) | Initial: 7.14% (n=1) | Initial: 7.14% (n=1) |
| | | | 48 | Subsequent: 77.08% (n=37) | Subsequent: 18.75% (n=9) | Subsequent: 4.17% (n=2) |
| | | Generic & biosimilar products | 8 | Initial: 87.50% (n=7) | Initial: 12.5% (n=1) | Initial: - |
| | | | 8 | Subsequent: 75.00% (n=6) | Subsequent: 12.50% (n=1) | Subsequent: 12.50% (n=1) |
| **Types of competing products** | P→S procurement group | Single competing product | 47 | 21.28% (n=10) | 63.83% (n=30) | 14.89% (n=7) |
| | | Mixed competing products | 26 | 46.15% (n=12) | 26.92% (n=7) | 26.92% (n=7) |
| | S→P procurement group | Single competing product | 55 | 80.00% (n=44) | 16.36% (n=9) | 3.64% (n=2) |
| | | Mixed competing products | 23 | 78.26% (n=18) | 13.04% (n=3) | 8.70% (n=2) |
| **Types of competing agents** | P→S procurement group | Single agent | 3 | 33.33% (n=1) | 33.33% (n=1) | 33.33% (n=1) |
| | | Mixed competing agents | 70 | 30.00% (n=21) | 51.43% (n=36) | 18.57% (n=13) |
| | S→P procurement group | Single agent | 4 | 75.00% (n=3) | 25.00% (n=1) | – |
| | | Mixed competing agents | 74 | 79.73% (n=59) | 14.86% (n=11) | 5.41% (n=4) |

particularly when a single bidder or supplier is participating in the tender process. This dynamic is not commonly permitted in segmented procurement systems, where prices are typically confined to the solicitation of quotations from multiple suppliers. Secondly, the substantial price decrease in the S→P group may be a result of suppliers being more amenable to offering competitive prices. Pooled procurers, given their inclination to make bulk purchases, may enjoy price breaks on larger orders. This willingness from suppliers to offer competitive prices is not solely driven by increased demand but rather by the advantageous nature of bulk purchases [17]. Thirdly, in terms of administrative costs, pooled procurement demonstrates potential cost reductions for both purchasers and suppliers. This efficiency arises from the streamlining of administrative processes and the reduction of bureaucracy associated with smaller-scale procurements. Contrastingly, within the P→S group, it was surprising to note a limited number of drugs experienced a decrease in prices. This unexpected observation suggests that specific drugs within this category could potentially derive advantages from this shift, underscoring the potential gains achievable through improved procurement efficiency. In a segmented procurement system, we posit that a higher number of suppliers might participate, facilitated by smaller supply sizes and more relaxed company requirements. Such a system is less constrained by stringent contract requirements, potentially fostering a more diverse participation of suppliers that stimulate heightened market competition which may reduce some of the drug prices. Additionally, suppliers in each market segment may exercise differing levels of bargaining power, cost structures, or market reach.

This study also revealed that prices tend to increase or remain the same when the procurement method changes. A transition from P→S appears to be significantly associated with the observed increase in purchase prices compared to the S→P group. The transition from P→S signifies a move from uniform pricing to the adoption of varied prices tailored for different customers or market segments. In this scenario, suppliers may be implementing market segmentation strategies for targeted pricing. Market segmentation involves the establishment of different prices for a product across various client segments or marketplaces, a practice known as price segmentation or discrimination [18,19]. Commercially, price discrimination is justified on grounds of representing cost savings and scalability [19]. Another contributing factor to the observed price increase from P→S could be the fragmentation of the supply chain, leading to inefficiencies and smaller volumes, elevated administrative costs, duplication, and coordination issues. Despite the likelihood of price increases, there is a tendency for prices to remain significantly more stable in P→S than S→P procurement. The findings show that segmented procurement is probably more adaptable to current prices, allowing for quicker adjustments to market shifts without significantly impacting prices. This study also observed that not all procurements that switched to pooled may offer price reductions. There was also a small percentage where the price would increase or stay the same despite the pooled initiative after switching from a segmented approach. If the price is increased in pooled procurement even with larger volumes, it may be more worthwhile to remain within the segmented procurement as it offers more flexibility to the buyer.

Further analyses shed additional light on the observed price dynamics resulting from the procurement changes among various types of products and competing product categories. In the P→S group, there are consistent trends among originators and reference products of biologics, showing similarities in the occurrences of price decreases, increases, and unchanged prices. On the other hand, the generic products in the P→S group have modest shifts, leaning towards the stability of the price changes than the originators and reference products of biologics. This may be attributed to the lower price band of generics. In the S→P procurement group, during the first tendering phase, the majority showcased a price decrease across all product types. This is supported by a theoretical model that assumes public purchasers are non-price-takers once purchasing in a centralized [17]. However, as subsequent tenders progressed, price reduction tended to slightly decrease, indicating a trend of diminishing price decreases after the initial tender. While collective procurement has significantly lowered drug prices in China, it may also increase the market power of winning manufacturers by securing multiple contracts across various products, potentially limiting competition [20]. This is contrary to the previous studies that highlighted that price reductions might be long-lasting because most tenders remained competitive over time [16]. This may be explained as the pooled procurement reduces the frequency of tendering, which may lead to reduced competition on the supply side, hence leading to shortages and increased drug prices in the long term [10]. In addition, the P→S group price increments were found to be more prevalent in a single-product scenario, while price reductions were more commonly observed when multiple competing products were involved. A study in China similarly found that pharmaceutical prices tend to decrease as the number of generic and therapeutic competitors increases, but interestingly, prices rose with a greater number of therapeutic classes [21]. In the United States, increased market competition is also associated with lower prices for generic topical drugs, with formulations having only 1–2 manufacturers experiencing significantly higher price increases compared to those with more competitors [22]. With an increased array of products available, more suppliers participate in the procurement process, competing to submit the most competitive bids. This fervent competition incentivizes suppliers to strive for lower prices, ultimately benefiting customers within a vigorously competitive market landscape.

Initially, firms maintain prices at the same level until market conditions change [23]. This dynamic pricing behavior emerges as suppliers adjust their prices in response to market states [24]. They aim to maintain market share through price adjustments while ensuring profitability through nuanced pricing strategies, all geared toward sustaining their position in the market. The product will be marketed as long as its marginal revenue exceeds its marginal cost to increase its profit [25]. Therefore, the ideal combination of product quality and pricing will most depend on the types of market to optimize earnings [25]. To avoid fierce price rivalry, however, and to be able to maintain prices above marginal costs for an extended period to maximize profits, competitors must differentiate their products [25].

## Policy and practice implications

The World Health Organization [26] has recognized procurement and pricing as the factors that enable access to medicines. In pursuit of greater pharmaceutical accessibility, government policies should prioritize purchasing medicines at the lowest possible costs while maintaining consistent minimum standards of quality. To achieve this, policy considerations should prioritize fostering competition among suppliers within both procurement systems to effectively drive prices down. However, only in a framework of fair competition is it possible to arrive at such a condition [27]. In a good procurement system purchasers should have enough information about demand and supply, market competition situation, and information on the price being offered to other health facilities. The government and public would benefit from getting the best value out of drug purchases once the knowledge gap between buyers is resolved. Perhaps, pricing transparency would be the first and most important step in achieving it [19].

Medicine security and self-reliance should be included as national policy for long term sustainability. In the Malaysian MOH, pooled procurement is used strategically to achieve national policy goals of increasing self-reliance by supporting local industries, which promotes security of supply, rather than increasing access to cheaper drugs. Other countries like Saudi Arabia, Algeria, Egypt, and Jordan incentivize local pharmaceutical manufacturing through procurement preferences [28]. Approaches in procurement preferences vary; for example, Algeria and Ghana allow domestic goods a price preference for procurement of up to 25% [28] and 15% [29], respectively, compared to imports. In addition, the public healthcare contracts usually have a term of two or three years in which besides cost savings, collaborative procurement may help to secure the supply chain and benefit both parties by sharing the risks posed by changes in drug prices or disruptions in supply. Contrary, segmented procurement does not provide the same level of risk-sharing in the event of supply disruptions or unexpected price changes as pooled procurement does.

Government policy should be designed with the strategic intent of seamlessly integrating both pooled and segmented procurement methods. This thoughtful integration aims to carefully assess and weigh the risks and benefits associated with each approach in the procurement process. Managing the pooled procurement process necessitates intricate coordination and significant resources [10]. Pooled procurement often provides a price advantage, but it limits customization and is not flexible enough to manage excessive demand under time pressure. Thorough planning is essential as over-reliance on a limited number of suppliers in pooled procurement can result in temporary disruptions to the supply chain. Alternatively, splitting contracted quantities among qualified suppliers mitigates shortage risks and promotes competitive bidding, as suppliers are incentivized to offer competitive terms to secure portions of the contract. Although prices are less advantage in segmented procurement, it has better respond to local demand, especially when urgent procurements are needed in response to emergencies such as disease outbreaks and/or adverse drug reactions following the use of certain products that are under the pooled procurement contract. In this type of procurement, the facility has full autonomy and control over supplier diversity, which may foster market competition related to pricing and/or service delivery. Moreover, segmented procurement can engage a different set of suppliers who may lack the financial eligibility, capacity, or motivation to participate in pooled procurement tenders.

On the other hand, the segmented procurement approach within the Malaysian MOH, subject to the set allowable threshold by the Ministry of Finance, indirectly emphasizes the significance of not disregarding the potential advantages in economies of scale and inventory security achievable through centralized competitive procurement. This stance finds support in a prior study that highlighted the superior performance of centralized and hybrid systems compared to decentralized systems across multiple factors. The study reported that centralized procurers roughly had 20% less expenditure while the hybrid procurers achieved approximately 9% lower expenditure than fully decentralized procurers [30].

To address the diverse landscape of procurement methods and ensure the efficient, cost-effective, and equitable procurement of medicines for public health systems, there is a need for the development of a dynamic and adaptable framework to guide the procurement process in the public healthcare sectors. This framework should be instrumental in navigating the complexities of the healthcare landscape, allowing for tailored strategies that align with the specific needs

of each health system by incorporating elements of pooled and segmented procurement, recognizing the potential benefits of both approaches. The framework must be meticulously crafted to embrace the inherent diversity in procurement practices, considering the distinctive requirements, challenges, and advantages linked with various procurement models. In the public healthcare sector, governance structures play a crucial role in overseeing procurement processes, ensuring compliance with regulatory standards, and managing budgets and supplier contracts with public accountability. A well-governed procurement system encourages transparency, where decisions are made objectively, fostering trust among suppliers and stakeholders. Additionally, it should incorporate considerations like economies of scale, inventory security, and supplier adaptability. This requires a strategic approach that harnesses the strengths of centralized competitive procurement while recognizing the flexibility and supplier diversity inherent in segmented procurement systems. Moreover, an ideal framework should also facilitate streamlined processes, minimize administrative burdens, and promote transparency and accountability across the entire procurement lifecycle.

## Strengths and limitations

This study is the first to examine the impact of drug pricing under both pooled and segmented procurement strategies within Malaysia's MOH public healthcare system. By exploring price dynamics in both directions of procurement switching, the study highlights the importance of context-specific decision-making in managing pharmaceutical procurement. While the findings offer important insights, several strengths and limitations must be acknowledged. First, the analysis focused on facilities with the highest drug expenditures. These facilities were purposively selected as they represent a significant share of MOH's overall procurement spending and are most likely to exhibit measurable pricing patterns relevant to large-scale procurement practices. While this focus enhances relevance, it may limit the generalizability of the findings to smaller or lower-volume facilities. Second, the median price was used to represent the outcome of segmented procurement. While this does not provide a volume-weighted average, the median was selected to reduce the influence of outliers and extreme price variations, offering a more stable measure of central tendency for price comparison purposes. Third, broader macroeconomic factors such as inflation, exchange rates, production costs, market concentration, and demand fluctuations were not included in this analysis. Due to the complexity and interdependence of these variables, isolating their individual effects was beyond the study's scope. This limitation has been explicitly acknowledged in the manuscript. Future research should incorporate these variables to develop a more comprehensive understanding of the drivers behind pharmaceutical pricing. Additionally, the switching effect between pooled and segmented procurement was analysed to observe price trends, but the observational nature of the study does not permit causal inferences. The analysis is also limited to selected therapeutic subgroups with high expenditure, and therefore, findings cannot be generalized to all pharmaceutical products. Despite these limitations, the study provides valuable preliminary insights into the relationship between procurement strategies and drug pricing. Future research should aim to evaluate the overall efficiency and effectiveness of the procurement process beyond pricing alone. Comparative assessments of procurement performance across various settings and procurement types may help identify opportunities to enhance the current framework. In particular, while pooled procurement is the dominant strategy, a deeper exploration into the role and efficiency of segmented procurement across diverse facility types could yield further strategies for optimizing procurement practices in Malaysia's public health system.

## Conclusion

Pooled procurement may present a superior approach to segmented procurement in reducing drug costs. However, integrating elements from both approaches proves more practical as both procurement strategies bring their own set of advantages and disadvantages, necessitating effective and efficient management strategies. A comprehensive understanding of the implications associated with transitioning between procurement types is critical. It enables policymakers to craft effective strategies that account for the enduring impact on healthcare budgets and overall system performance.

## Supporting information

**S1 Appendix: The details of price changes in the P→S and S→P procurement groups.**
(XLSX)

## Acknowledgments

We would like to thank the Director General of the Malaysian MOH for permission to publish this article. We would like to express our special thanks to the Pharmaceutical Services Programme, particularly the Pharmacy Practice & Development Division and the Pharmacy Policy & Strategic Planning Division for enabling and providing the necessary resources to conduct this study. We extend our heartfelt gratitude to the Medicines Price Management Branch, Pharmaceutical Logistic Management Branch, the State Health Departments, and the 19 MOH facilities for their permission and assistance in collecting the data.

## Author contributions

**Conceptualization:** Farahwahida Mohd Kasim, Ernieda Hatah, Lokhman Hakim Osman, Adliah Mhd Ali, Zaheer-Ud-Din Babar.

**Data curation:** Farahwahida Mohd Kasim.

**Formal analysis:** Farahwahida Mohd Kasim.

**Funding acquisition:** Ernieda Hatah.

**Investigation:** Farahwahida Mohd Kasim.

**Methodology:** Farahwahida Mohd Kasim, Ernieda Hatah, Lokhman Hakim Osman, Adliah Mhd Ali, Zaheer-Ud-Din Babar.

**Project administration:** Farahwahida Mohd Kasim, Ernieda Hatah.

**Software:** Farahwahida Mohd Kasim.

**Supervision:** Ernieda Hatah, Lokhman Hakim Osman, Adliah Mhd Ali, Zaheer-Ud-Din Babar.

**Validation:** Ernieda Hatah, Lokhman Hakim Osman, Adliah Mhd Ali, Zaheer-Ud-Din Babar.

**Visualization:** Farahwahida Mohd Kasim.

**Writing – original draft:** Farahwahida Mohd Kasim.

**Writing – review & editing:** Farahwahida Mohd Kasim, Ernieda Hatah, Lokhman Hakim Osman, Adliah Mhd Ali, Zaheer-Ud-Din Babar.

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
