## [Decision Letter · Decision Letter 0]

11 Oct 2024

Dear Dr. Hatah,

Thank you for submitting your manuscript to PLOS ONE. After careful consideration, we feel that it has merit but does not fully meet PLOS ONE’s publication criteria as it currently stands. Therefore, we invite you to submit a revised version of the manuscript that addresses the points raised during the review process.

We look forward to receiving your revised manuscript.

Kind regards,

Madhavi Yennapu, PhD

Academic Editor

PLOS ONE

Journal Requirements: When submitting your revision, we need you to address these additional requirements. 1. Please ensure that your manuscript meets PLOS ONE's style requirements, including those for file naming. The PLOS ONE style templates can be found at https://journals.plos.org/plosone/s/file?id=wjVg/PLOSOne_formatting_sample_main_body.pdf and https://journals.plos.org/plosone/s/file?id=ba62/PLOSOne_formatting_sample_title_authors_affiliations.pdf 2. Thank you for stating the following in the Acknowledgments Section of your manuscript: "The authors would like to acknowledge the financial support from the Malaysian Ministry of Higher Education. We also thank the Director General of the Malaysian Ministry of Health for permission to publish this article. We would like to express our special thanks to the Pharmaceutical Services Programme, particularly the Pharmacy Practice & Development Division, the Pharmacy Policy & Strategic Planning Division, and the NPRA for enabling and providing the necessary resources to conduct this study. We extend our heartfelt gratitude to the Medicines Price Management Branch, Pharmaceutical Logistic Management Branch, the State Health Departments, and the 19 MOH facilities for their permission and assistance in collecting the data." We note that you have provided funding information that is not currently declared in your Funding Statement. However, funding information should not appear in the Acknowledgments section or other areas of your manuscript. We will only publish funding information present in the Funding Statement section of the online submission form. Please remove any funding-related text from the manuscript and let us know how you would like to update your Funding Statement. Currently, your Funding Statement reads as follows: "This study was funded by the Fundamental Research Grants Scheme of the Ministry of Higher Education of Malaysia. Grant number FRGS/1/2022/SS02/UKM/02/1 was assigned, and EH received this funding. The funder had no role in the study design, data collection and analysis, decision to publish, or preparation of the manuscript. Funder website:  Pleasehttps://mastic.mosti.gov.my/sti/incentives/fundamental-research-grant-scheme-frgs include your amended statements within your cover letter; we will change the online submission form on your behalf. 3. We note that you have indicated that there are restrictions to data sharing for this study. For studies involving human research participant data or other sensitive data, we encourage authors to share de-identified or anonymized data. However, when data cannot be publicly shared for ethical reasons, we allow authors to make their data sets available upon request. For information on unacceptable data access restrictions, please see .  Beforehttp://journals.plos.org/plosone/s/data-availability#loc-unacceptable-data-access-restrictions we proceed with your manuscript, please address the following prompts: a) If there are ethical or legal restrictions on sharing a de-identified data set, please explain them in detail (e.g., data contain potentially identifying or sensitive patient information, data are owned by a third-party organization, etc.) and who has imposed them (e.g., a Research Ethics Committee or Institutional Review Board, etc.). Please also provide contact information for a data access committee, ethics committee, or other institutional body to which data requests may be sent. b) If there are no restrictions, please upload the minimal anonymized data set necessary to replicate your study findings to a stable, public repository and provide us with the relevant URLs, DOIs, or accession numbers. Please see http://www.bmj.com/content/340/bmj.c181.long for guidelines on how to de-identify and prepare clinical data for publication. For a list of recommended repositories, please see https://journals.plos.org/plosone/s/recommended-repositories. You also have the option of uploading the data as Supporting Information files, but we would recommend depositing data directly to a data repository if possible. Please update your Data Availability statement in the submission form accordingly.

**Additional Editor Comments:**

Both the reviewers found the paper interesting and feel that it deserves publication in PLOS1.

Reviewer-1 recommends ‘accept’ without any critical comments and suggestions.

Reviewer-2 recommends ‘major revision’. Reviewer 2 has given constructive critical comments/queries and suggestions to enrich this submission that clearly substantiates evidence with respect to the analysis of results and the resultant discussion and conclusions. Reviewer's comments are enclosed in a separate sheet.

Academic Editor’s View: The title of the article is interesting, important and relevant with respect to access to affordable medicines if one understands the procurement strategies/system. The following are the observations in the current form of the manuscript.

i) The broad message of the article seemed reasonable. The manuscript has clarity in its introduction and discussion (partly), but the presentation of data collection and analysis are unclear.

ii) The methodology seems quite complex! The way it is presented may need attention.

iii) Though the statistical analysis was carried out appropriately (acknowledged by both the reviewers), there is no enough explanation provided by the authors exactly what protocols/steps followed in the methodology and how it was done. Stating the limitations of methodology and results analysis within the respective sections may help.

iv) Some terms used are unfamiliar and unclear to all readers. For instance. why “Lamparian q” is called so, and it’s not clear what is the difference between “Lamparian Q” and “direct purchase”. Under what situations/conditions, “Lamparian Q” or “direct purchase” are used/opted?

v) The discussion is interesting provided a little more clarity in its explanation. Incorporation of suggestions of Reviewer-2 may be useful.

vi) Suggest the authors to address the queries pointed out by the Reviewer-2, incorporate his/her suggestions and resubmit the article.

Reviewers' comments:

Reviewer's Responses to Questions

**Comments to the Author**

1. Is the manuscript technically sound, and do the data support the conclusions?

Reviewer #1: Yes

Reviewer #2: Partly

2. Has the statistical analysis been performed appropriately and rigorously?

Reviewer #1: Yes

Reviewer #2: Yes

3. Have the authors made all data underlying the findings in their manuscript fully available?

Reviewer #1: No

Reviewer #2: No

4. Is the manuscript presented in an intelligible fashion and written in standard English?

Reviewer #1: Yes

Reviewer #2: Yes

Reviewer #1: I reviewed the work and thought it fascinating. It was a well-written manuscript. The introduction was well justified, and the methodology was sound and well-planned. The outcome was presented and described appropriately. The discussion went well.

Reviewer #2: The article is interesting and on an important topic. It could be a valuable contribution to the literature, particularly to provide analysis of medicine procurement in Malaysia and inform the broader field. The writing is technical and dense, which makes it hard to read particularly because the underlying data is restricted and important details are missing. Analyses could be sharpened significantly (especially small ns and within sub-group analysis) and the discussion should be supported by results. More detailed comments are in the attachment.

**Do you want your identity to be public for this peer review?** For information about this choice, including consent withdrawal, please see our Privacy Policy

Reviewer #1: No

Reviewer #2: **Yes: ** Malini Aisola

---

## [Author Response · Author response to Decision Letter 1]

17 Dec 2024

Please refer to the responses addressing specific reviewer and editor comments in the attached file.

---

## [Decision Letter · Decision Letter 1]

24 Apr 2025

Dear Dr. Hatah,

Thank you for submitting your manuscript to PLOS ONE. After careful consideration, we feel that it has merit but does not fully meet PLOS ONE’s publication criteria as it currently stands. Therefore, we invite you to submit a revised version of the manuscript that addresses the points raised during the review process.

We look forward to receiving your revised manuscript.

Kind regards,

Jianhong Zhou

Staff Editor

PLOS ONE

Reviewers' comments:

Reviewer's Responses to Questions

**Comments to the Author**

Reviewer #1: All comments have been addressed

Reviewer #3: All comments have been addressed

Reviewer #4: (No Response)

2. Is the manuscript technically sound, and do the data support the conclusions?

Reviewer #1: Yes

Reviewer #3: Yes

Reviewer #4: Partly

3. Has the statistical analysis been performed appropriately and rigorously?

Reviewer #1: Yes

Reviewer #3: Yes

Reviewer #4: Yes

4. Have the authors made all data underlying the findings in their manuscript fully available?

Reviewer #1: Yes

Reviewer #3: (No Response)

Reviewer #4: Yes

5. Is the manuscript presented in an intelligible fashion and written in standard English?

Reviewer #1: Yes

Reviewer #3: Yes

Reviewer #4: Yes

Reviewer #1: I have reviewed the revised version of the manuscript and find it significantly improved. I am pleased to recommend it for publication.

Reviewer #3: (No Response)

Reviewer #4: Introduction:

1. Phrasing such as "In the study prices of medicines were significantly higher..." is vague. Which study? It needs to be properly cited.

2. The sentence "the study reveals a concession in Malaysia's public sector..." is grammatically and contextually unclear. “Concession" is a vague term here please clarify the meaning.

3. In the sentence, "Thus, efficient, and cost-effective expenditure is crucial..." “Expenditure" is not the right word in this context; perhaps "procurement strategies" or "cost control" would be more optimal.

4. Phrases like "organizations can strategically align their efforts to address the escalating drug spending..." are repeated at the beginning and end.

5. The terms "collaborative" and "non-collaborative" procurement strategies are introduced abruptly. A brief definition or example before listing the types would improve understanding.

6. Pooled procurement is mentioned multiple times without a consistent definition, and a required definition to improve readers' understanding.

7. There is no explanation about “how procurement is currently structured in Malaysia’s public healthcare setting.

8. It would be better if the authors first discuss international trends, Malaysia’s spending, procurement models, and gaps in research.

Material & Methods

1. As an observational and retrospective study, causality cannot be established, only associations or trends can be

2. The purposive sampling strategy may introduce selection bias, as it doesn't ensure representation of all drug classes or procurement practices.

3. Only drugs with expenditure ≥ RM4 million were included, which may exclude low-volume/high-cost drugs or essential but rarely used medications.

4. The exclusion of APPL/concession drugs may miss many commonly used medications, affecting price trend analysis

5. No mention of data validation methods or handling of missing/inconsistent data, which could impact data reliability.

6. It is unclear whether the same brands/manufacturers were compared between pooled and segmented procurement methods, which may affect the results.

7. External market factors (e.g., currency fluctuations, raw material costs, inflation) are not controlled or adjusted for, which could affect procurement prices independently of the method.

8. Include a flow diagram or table summarizing data inclusion/exclusion to enhance transparency.

**Do you want your identity to be public for this peer review?** For information about this choice, including consent withdrawal, please see our Privacy Policy

Reviewer #1: No

Reviewer #3: No

Reviewer #4: **Yes: ** Dr. Dinesh Kumar Meena

---

## [Author Response · Author response to Decision Letter 2]

9 Jun 2025

Dear reviewers,

Thank you for your comments. We have made the necessary explanation and improved the introduction, material and methods in accordance with the specific comments provided by the reviewer. Please refer to the second response to reviewers and the revised article with the tracked changes in the attached file.

---

## [Decision Letter · Decision Letter 2]

4 Sep 2025

Dear Dr. Hatah,

Thank you for submitting your manuscript to PLOS ONE. After careful consideration, we feel that it has merit but does not fully meet PLOS ONE’s publication criteria as it currently stands. Therefore, we invite you to submit a revised version of the manuscript that addresses the points raised during the review process.

We look forward to receiving your revised manuscript.

Kind regards,

Rabia Hussain, PhD

Academic Editor

PLOS ONE

Journal Requirements:

Additional Editor Comments:

**The manuscript needs some minor adjustments, so kindly address the comments.**

Reviewers' comments:

Reviewer's Responses to Questions

**Comments to the Author**

Reviewer #5: All comments have been addressed

Reviewer #6: All comments have been addressed

2. Is the manuscript technically sound, and do the data support the conclusions?

Reviewer #5: Yes

Reviewer #6: Yes

3. Has the statistical analysis been performed appropriately and rigorously?

Reviewer #5: Yes

Reviewer #6: Yes

4. Have the authors made all data underlying the findings in their manuscript fully available?

Reviewer #5: Yes

Reviewer #6: Yes

5. Is the manuscript presented in an intelligible fashion and written in standard English?

Reviewer #5: Yes

Reviewer #6: Yes

Reviewer #5: Many thanks for addressing all the comments. Just one minor change is needed.

Please change the heading Conclusions to Conclusion (singular term).

Reviewer #6: The abstract provides a comprehensive overview but could benefit from clearer delineation of the study's objectives and key findings.

The discussion could be strengthened by including a brief comparison with similar studies in other countries to highlight broader implications.

**Do you want your identity to be public for this peer review?** For information about this choice, including consent withdrawal, please see our Privacy Policy

Reviewer #5: No

Reviewer #6: **Yes: ** Nour Aymn Ahmad

---

## [Author Response · Author response to Decision Letter 3]

9 Sep 2025

Reviewer #5: Many thanks for addressing all the comments. Just one minor change is needed. Please change the heading Conclusions to Conclusion (singular term).

Answer: Thank you for your kind feedback and for highlighting the needed correction. The heading has now been changed from Conclusions to Conclusion as requested. (page 29, line 623)

Reviewer #6: The abstract provides a comprehensive overview but could benefit from clearer delineation of the study's objectives and key findings. The discussion could be strengthened by including a brief comparison with similar studies in other countries to highlight broader implications.

Answer: We appreciate your suggestion to improve the clarity of the abstract. The revised abstract now includes a clearer delineation of the study's objective and key findings to enhance readability and better reflect the study’s contribution. Specifically, we have clearly stated the aim upfront, followed by a concise summary of methods and highlighted the main statistical findings and implications. (page 2, lines 24-42) In response to your second comment, we have expanded the Discussion section to include a brief comparison with similar studies conducted in other countries, such as China, and the United States. These comparisons serve to contextualize our findings within broader health system settings. (page 23, lines 483-485 & 492-497)

---

## [Decision Letter · Decision Letter 3]

21 Dec 2025

Drug price dynamics following changes in procurement method in the public healthcare setting in Malaysia.

PONE-D-24-09587R3

Dear Dr. Hatah,

We’re pleased to inform you that your manuscript has been judged scientifically suitable for publication and will be formally accepted for publication once it meets all outstanding technical requirements.

Kind regards,

Charles C Ezenduka, PhD

Academic Editor

PLOS One

Additional Editor Comments (optional):

Reviewers' comments:

Reviewer's Responses to Questions

**Comments to the Author**

Reviewer #6: All comments have been addressed

2. Is the manuscript technically sound, and do the data support the conclusions?

Reviewer #6: Yes

3. Has the statistical analysis been performed appropriately and rigorously?

Reviewer #6: Yes

4. Have the authors made all data underlying the findings in their manuscript fully available?

Reviewer #6: Yes

5. Is the manuscript presented in an intelligible fashion and written in standard English?

Reviewer #6: Yes

Reviewer #6: I have no comments for the authors. The manuscript is clear, well structured, and requires no revisions

**Do you want your identity to be public for this peer review?** For information about this choice, including consent withdrawal, please see our Privacy Policy

Reviewer #6: **Yes: ** Nour Aymn Ahmad

---

## [Editor Report · Acceptance letter]

PONE-D-24-09587R3

PLOS One

Dear Dr. Hatah,

I'm pleased to inform you that your manuscript has been deemed suitable for publication in PLOS One. Congratulations! Your manuscript is now being handed over to our production team.

Kind regards,

on behalf of

Dr. Charles C Ezenduka

Academic Editor

PLOS One